# Quantitative, Targeted Analysis of Gut Microbiota Derived Metabolites Provides Novel Biomarkers of Early Diabetic Kidney Disease in Type 2 Diabetes Mellitus Patients

**DOI:** 10.3390/biom13071086

**Published:** 2023-07-07

**Authors:** Lavinia Balint, Carmen Socaciu, Andreea Iulia Socaciu, Adrian Vlad, Florica Gadalean, Flaviu Bob, Oana Milas, Octavian Marius Cretu, Anca Suteanu-Simulescu, Mihaela Glavan, Silvia Ienciu, Maria Mogos, Dragos Catalin Jianu, Ligia Petrica

**Affiliations:** 1Department of Internal Medicine II—Division of Nephrology, “Victor Babes” University of Medicine and Pharmacy, Eftimie Murgu Sq. No. 2, County Emergency Hospital, 300041 Timisoara, Romania; lavinia.balint@umft.ro (L.B.); flaviu_bob@yahoo.com (F.B.); oana.milas@yahoo.com (O.M.); anca.simulescu@yahoo.com (A.S.-S.); mihaelapatruica@gmail.com (M.G.); ienciu_silviaoana@yahoo.com (S.I.); maria.stefan2014@yahoo.com (M.M.); ligia_petrica@yahoo.co.uk (L.P.); 2Center for Molecular Research in Nephrology and Vascular Disease, Faculty of Medicine, “Victor Babes” University of Medicine and Pharmacy, Eftimie Murgu Sq. No. 2, 300041 Timisoara, Romania; csocaciudac@gmail.com (C.S.); vlad.adrian@umft.ro (A.V.); jianu.dragos@umft.ro (D.C.J.); 3Research Center for Applied Biotechnology and Molecular Therapy Biodiatech, SC Proplanta, Trifoiului 12G, 400478 Cluj-Napoca, Romania; 4Department of Occupational Health, University of Medicine and Pharmacy “Iuliu Haţieganu”, Victor Babes 8, 400347 Cluj-Napoca, Romania; andreeaiso@gmail.com; 5Department of Internal Medicine II—Division of Diabetes and Metabolic Diseases, “Victor Babes” University of Medicine and Pharmacy, Eftimie Murgu Sq. No. 2, County Emergency Hospital, 300041 Timisoara, Romania; 6Department of Surgery I—Division of Surgical Semiology I, “Victor Babes” University of Medicine and Pharmacy, Eftimie Murgu Sq. No. 2, Emergency Clinical Municipal Hospital, 300041 Timisoara, Romania; tavicretu@yahoo.com; 7Department of Neurosciences—Division of Neurology, “Victor Babes” University of Medicine and Pharmacy, Eftimie Murgu Sq. No. 2, County Emergency Hospital, 300041 Timisoara, Romania; 8Center for Cognitive Research in Neuropsychiatric Pathology (Neuropsy-Cog), Faculty of Medicine, “Victor Babes” University of Medicine and Pharmacy, Eftimie Murgu Sq. No. 2, 300041 Timisoara, Romania; 9Center for Translational Research and Systems Medicine, Faculty of Medicine, “Victor Babes” University of Medicine and Pharmacy, Eftimie, Murgu Sq. No. 2, 300041 Timisoara, Romania

**Keywords:** diabetic kidney disease, serum biomarkers, urine biomarkers, amino acids, uremic toxins, carnitines, targeted metabolomics

## Abstract

Diabetic kidney disease (DKD) is one of the most debilitating complications of type 2 diabetes mellitus (T2DM), as it progresses silently to end-stage renal disease (ESRD). The discovery of novel biomarkers of early DKD becomes acute, as its incidence is reaching catastrophic proportions. Our study aimed to quantify previously identified metabolites from serum and urine through untargeted ultra-high-performance liquid chromatography coupled with electrospray ionization-quadrupole-time of flight-mass spectrometry (UHPLC-QTOF-ESI+-MS) techniques, such as the following: arginine, dimethylarginine, hippuric acid, indoxyl sulfate, p-cresyl sulfate, L-acetylcarnitine, butenoylcarnitine and sorbitol. The study concept was based on the targeted analysis of selected metabolites, using the serum and urine of 20 healthy subjects and 90 T2DM patients with DKD in different stages (normoalbuminuria—uACR < 30 mg/g; microalbuminuria—uACR 30–300 mg/g; macroalbuminuria—uACR > 300 mg/g). The quantitative evaluation of metabolites was performed with pure standards, followed by the validation methods such as the limit of detection (LOD) and the limit of quantification (LOQ). The following metabolites from this study resulted as possible biomarkers of early DKD: in serum—arginine, dimethylarginine, hippuric acid, indoxyl sulfate, butenoylcarnitine and sorbitol and in urine—p-cresyl sulfate.

## 1. Introduction

Type 2 diabetes mellitus (T2DM) is a terrible condition, encountered at the crossroad of various systemic complications regarding the renal, cerebro- and cardio-vascular systems. The prevalence of T2DM is continuously expanding, with 415 million cases reported in the world in 2015, a number that is foreseen to grow by 10.4% by 2040. Hence, diabetic kidney disease (DKD), the most frequent complication, is directly proportional to the prevalence of T2DM and has a significant impact on the quality of life, representing a serious economic burden [1].

Previously entitled “the silent killer”, DKD develops insidiously and progresses inevitably to end-stage renal disease (ESRD), in the absence of an early diagnosis and prompt therapeutical intervention [2]. The reduction of glomerular filtration rate or the occurrence of albuminuria represent criteria by which DKD diagnosis is defined, but these are imperfect indicators of renal damage [3]. By the time DKD becomes biologically evident, the structural and functional renal lesions are already contouring chronic features, this process becoming only slightly reversible or completely irreversible [4,5].

To date, metabolic disbalances due to T2DM that occur before the onset of albuminuria remain insufficiently explored. Metabolomics is a comprehensive analysis method which takes part of the “omics” sciences along with transcriptomics, proteomics, lipidomics, etc. [6] Metabolomic methods are based on analyses which can be untargeted (the identification and discrimination of all metabolites) and targeted (quantification of the selected metabolites); these analyses are performed on samples which may include human tissues, as well as biofluids. The metabolites obtained from these techniques, expressed in molecular mass/retention times, are confronted with information from the human metabolome database (HMDB) for their specific identification [7]. They may be influenced by individual factors (diet, physical activity and gut microbiota) and environmental factors (lifestyle and pollution). Hence, the metabolites identified express a pattern that may be equivalent to the uniqueness of a fingerprint, providing a personalized profile regarding the diseases and drug interferences of a certain individual [8].

The gut is a giant reservoir of microbiota-derived metabolites. Diabetes mellitus itself produces a metabolic shift in metabolite production and reabsorption in the gut. This process, driven by hyperglycemia, is enhanced by metabolite accumulation in the context of renal impairment [9,10]. Thus, the research directed through gut microbiota-derived metabolites and their dynamics may represent a turning point over the actual perception of T2DM pathophysiologic processes.

Metabolites belonging to the free amino acids class, such as arginine and its methylated amines, seem to have an interesting interplay in renal disease. Some authors have found low levels of arginine and high levels of asymmetric dimethylarginine (ADMA) in the serum of patients suffering from chronic kidney disease [11]. Going further, Jayachandran et al. discovered an ascending trend of ADMA levels from T2DM without DKD to micro- and macroalbuminuria [12].

Uremic toxins are molecules that accumulate once the renal clearance declines. Gut microbiota is a major source of hippuric acid, indoxyl sulfate and p-cresyl sulfate, according to literature data [13]. These metabolites are capable of disrupting the normal homeostasis by having an enhanced biological activity in certain organs [14].

Hippuric acid is a protein-bound metabolite that either forms due to mitochondrial glycation of benzoic acid in the liver or kidney, or this derives from phenylalanine metabolism under the action of gut bacteria [15]. Fang et al. point out decreased serum levels of hippuric acid, whereas Sharma et al. describe lower urinary levels of this metabolite when comparing DKD patients to controls [16,17]. In contrast, hippurate was found to be increased in urine in DKD patients by Li et al. [18].

Based on anterior studies, the levels of indoxyl sulfate were found to be increased in serum and in urine in patients suffering from DKD [19,20]. Additionally, the levels of p-cresyl sulfate were found to depend on serum creatinine levels [21]. Few studies have emerged with regard to its urinary concentrations in certain stages of DKD.

Carnitines are dietary or endogenous compounds, synthetized in many organs, including in the gut and kidney and filtered by the kidney and reabsorbed in proportion of 90%. Their role consists of the transport of fatty acid to mitochondria in order to produce energy [22]. Acylcarnitines are esterified carnitines that are the best predictors of well-functioning mitochondria. Van der Kloet et al. found out that high levels of urinary acylcarnitines are involved in incipient DKD pathogenesis [23]. Moreover, Aichler et al. discovered high levels of acylcarnitines in serum of T2DM mice, with implication in β cell dysfunction and subsequent derangements in insulin secretion [24].

Sorbitol derives from glucose under the aldose reductase activity, reflecting the activation of polyol pathway. The diabetic state enhances the production of sorbitol, with subsequent production of advance glycation end-products (AGEs). Sorbitol was found to be increased in a study performed in T2DM patients with obesity [25]. In addition, Valdes et al. found increased concentrations of sorbitol in cultures of tubular proximal human cells [26].

Unravelling the incipient pathological modifications in DKD is becoming an urgent global issue, as ESRD incidence related to DKD continuously expands, rapidly reaching epidemic proportions. This study arises as a sequel of the untargeted multivariate and univariate metabolomic analyses of serum and urine in T2DM patients with DKD [27] and it aims to describe the subsequent targeted analysis of the metabolites that expressed a strong biomarker potential.

## 2. Materials and Methods

### 2.1. Selection of Study Participants and Ethical Principles

Our study encompassed the screening of 130 consecutive TD2M patients, from the Department of Nephrology and the Ambulatory of Nephrology, and from the Department of Diabetes and Metabolic Diseases and the Ambulatory of Diabetes and Metabolic Diseases, County Emergency Hospital Timişoara. Briefly, the inclusion criteria consisted of the enrollment of patients with a DM duration more than 5 years and HbA1c < 10%. Patients excluded were those who were suffering from non-diabetic kidney disease (NDKD) of all etiologies, end-stage renal disease, ongoing infections, neoplasia, autoimmune diseases, psychiatric disorders and those presenting with uncontrolled DM (HbA1c ≥ 10%).

In the end, patients’ screening provided 90 eligible T2DM (P) patients based on the inclusion criteria and 20, additional, control subjects (C) which were not suffering of any acute or chronic illness, found in the general physician’s records, as described in our previous study [27]. The T2DM group was divided into three subgroups, with respect to KDIGO Guidelines [P1—normoalbuminuria: uACR < 30 mg/g; P2—microalbuminuria: uACR ≥ 30 < 300 mg/g, macroalbuminuria—P3: uACR ≥ 300 mg/g] [28]. Groups P and C were age and gender matched and all of the patients in the DKD group were under treatment with oral antidiabetic agents and/or insulin, angiotensin 2 converting enzyme inhibitors/angiotensin 2 receptor blockers, and statins. The clinical and biological parameters of the participants are described in Table 1.

The Scientific Research Ethics Committees from County Emergency Hospital Timişoara (220/18 January 2021) and “Victor Babes” University of Medicine and Pharmacy Timişoara (29/30 June 2021) approved this study. All participants signed an informed consent document. All procedures were conducted according to the criteria set by the Declaration of Helsinki.

### 2.2. Chemicals and Reagents

The chemicals and reagents used in this study were acquired as follows: PLC-grade formic acid from Sigma-Aldrich (Burlington, MA, USA)], HPLC/MS-grade formic acid and acetonitrile from Fisher Scientific (Pittsburg, PA, USA). The pure standard biomarkers that were utilized are represented by acetyl-L-carnitine hydrochloride (J6153606; Alfa Aesar by Thermo Fisher) MW = 203; arginine from Amino acid standard H (product #20088, Thermo Scientific) MW = 174; asymmetric dimethyl-L-arginine ≥ 95% (HPLC) CAS (Thermo Scientific) 30315-93-6 (Sigma Aldrich) MW = 202.25; hippuric acid, 98%, (A1269022; Alfa Aesar by Thermo Fisher) MW = 179; indoxyl sulfate potassium salt, 97%, (A1707901; Alfa Aesar by Thermo Fisher) MW = 213; Sorbitol, >98% product S1876 Sigma-Aldrich Chemie GmbH, MW = 182; p-Cresyl sulfate, >98%, product 29504 Cayman Chemical, US, MW = 188 and creatinine > 98% product C4255, Sigma-Aldrich Chemie GmbH, MW = 113. As an internal standard Doxorubicin hydrochloride was used (MW = 580) (injectable, 2 mg/mL Sun Pharmaceutical Industries). Ultra-high purity water was prepared by Millipore-Q Water Purification System (Millipore, Germany). LC–MS grade MeOH, MeCN, and formic acid were purchased from Fisher Scientific (Loughborough, UK). Ultra-pure water was purified by a Milli-Q water system (Millipore, Milford, MA, USA). Instruments used in this study included a vortex mixer, Minicentrifuge Eppendorf (Thermo Fisher Scientific, USA), UPLC-Q-TOF/MS (Bruker GmbH, Bremen, Germany).

### 2.3. Sample Preparation

The blood was collected by venipuncture in sterile vacutainers without anticoagulant, and the serum was kept at −80 °C until analysis. They were labeled using confidential numerical codes. The urine samples were collected in the morning in sterile vials. A volume of 0.8 mL mix of pure HPLC-grade Methanol and Acetonitrile (2:1 *v*/*v*) was added for each volume of 0.2 mL of blood serum and 0.2 mL of urine, respectively. In each case the mixture was vortexed to precipitate proteins, ultrasonicated for 5 min and kept for 24 h at −20 °C for increasing the protein precipitation. The supernatant was collected after centrifugation at 12,500 rpm for 10 min (4 °C) and filtered through nylon filters (0.2 μm). Finally, it was placed in glass micro vials and introduced in the autosampler of the ultra-high-performance liquid chromatograph (UHPLC) before injection. The supernatant was transferred to an autosampler vial for HPLC-MS analysis. Quality control (QC) samples from a mix of 0.1 mL from each serum or urine samples were also obtained and used as representative generic samples which were injected at the beginning and end and as every 10th injection while analyzing the study samples.

### 2.4. UHPLC-QTOF-ESI^+^-MS Analysis

The metabolomic profiling was performed by UHPLC-QTOF-ESI+-MS, using a ThermoFisher Scientific UHPLC Ultimate 3000 instrument equipped with a quaternary pump, Dionex delivery syste, and MS detection equipment with MaXis Impact (Bruker Daltonics, Waltham, MA, USA). The metabolites were separated on an Acclaim C18 column (5 μm, 2.1 × 100 mm, pore size of 30 nm) (Thermo Scientific) at 28 °C. The mobile phase consisted of 0.1% formic acid in water (A) and 0.1% formic acid in acetonitrile (B). The elution time was set for 20 min (min). The flow rate was set at 0.3 mL · min^−1^ for serum samples and 0.8 mL · min^−1^ for urine samples. The gradient for serum samples was 90 to 85% A (0–3 min), 85–50% A (3–6 min), 50–30% (6–8 min), 30–5% (8–12 min) and afterward increased to 90% at min 20. The gradient for urine samples was 90 to 85% A (0–3 min), 85–30% A (3–6 min), 30–10% (6–8 min), isocratic until min 12 and then increased until 90% at min 20. The volume of injected extract was 5 µL, the column temperature was 25 °C. Two categories of QC samples were obtained by mixing similar volumes of blood serum and urine, respectively, in order to calibrate the separations. Doxorubicin hydrochloride (*m*/*z* = 581.3209) (2 mg/mL) was also added to QC samples as an internal standard (IS).

The MS parameters were ionization mode positive (ESI+), MS calibration with Natrium formate, capillary voltage 3500 V, the pressure for the nebulizing gas 2.8 barr, drying gas flow 12 L/min and drying temperature 300 °C. The *m*/*z* values to be separated were set between 60 and 600 Daltons. The control of the instrument and the data processing was done using the specific software TofControl 3.2, Chromeleon, and HyStar 3.2, Data Analysis 4.2, respectively (Bruker Daltonics).

### 2.5. Data Processing and Statistical Analysis

Initially, the untargeted analysis was performed for the same groups (P and C). As a first step in the discrimination between the two groups, multivariate analysis was performed which included Fold Change, Principal Component Analysis (PCA) and Partial Least Squares-Discriminant Analysis (PLSDA) score plots, including VIP values. Log2 Fold Change values and Bonferroni-adjusted *p*-values were used for the generation of volcano plots. The statistical significance was defined at *p*-value under 0.05. In order to test the discriminatory capacity of each metabolite, Receiver Operating Characteristic (ROC) analysis was performed. Area Under Curve with values ≥0.8 displayed a very high prediction effect of identified metabolites on disease, namely, that these metabolites are candidate biomarkers of further studies. In a second step, the one-way ANOVA univariate analysis aimed to discriminate controls (C) from the subgroups of patients (P1, P2, P3). The PCA and PLSDA score plots including VIP values, cross-validation parameters, as well as the Mean Decrease Accuracy (MDA) scores by Random Forest analysis were performed. For the statistical analysis of targeted metabolites matrices were selected which included the five metabolites mentioned above (*m*/*z* values vs. MS peak intensity, as .csv file) [27].

Subsequently, the data obtained from untargeted metabolomic analysis provided a number of seven specific molecules that were chosen to be targeted in plasma and urine. Mass spectrometry (MS) peak intensities of these molecules were considered for the comparison of the established subgroups (C, P1, P2, P3). In addition, the mean values of peak intensity (PI) and their standard deviations (SD) were calculated for these specific biomarkers in agreement with recent literature data. For quantitative analysis, the calibration curves were built with pure standards.

### 2.6. Metabolites Identification

The data obtained by the untargeted metabolic fingerprint of samples provided seven metabolites in serum and seven metabolites in urine to the targeted analysis. These metabolites expressed statistical significance as putative biomarkers of differentiation between groups, based on MS peak intensities. The matrices which included these seven metabolites were selected for the statistical analysis, using the Metaboanalyst 5.0 platform for multivariate and univariate analysis (https://www.metaboanalyst.ca, accessed on 29 April 2023). This platform was also utilized for exploring how the major metabolic pathways related to the differential metabolites were affected.

The human metabolome database HMDB (http://www.hmdb.ca, accessed on 29 April 2023), PubChem (https://pubchem.ncbi.nlm.nih.gov, accessed on 29 April 2023) and Lipidmaps (https://www.lipidmaps.org, accessed on 29 April 2023), provided the chemical information about metabolites, considering a deviation of the *m*/*z* value of 0.05. The identification results were strengthened by the combination of the exact number (*m*/*z*) (without exceeding 0.05 Da) with an ionization method that meets the experimental condition. Furthermore, the results were proved by the comparison of primary and secondary mass spectra information of the differential metabolites and the theoretical fragments of HMDB search results (https://www/hmdb.co, accessed on 29 April 2023). For quantitative analysis, the calibration curves were built with pure standards.

### 2.7. Quantitative Evaluation

To prepare the calibration solution and quality control (QC) samples, the following stock solutions of the potential biomarkers were used: Creatinine 2 mM, Arginine 1 mM, ADMA 1 mM, L-Acetylcarnitine 1 mM, Hippuric acid 1mM, sorbitol 1 mM, Indoxyl sulfate 1 mM, p-Cresyl sulfate 1 mM, dissolved in ultra-pure water and/or methanol. As an internal standard Doxorubicin hydrochloride (DOXO) was used at a stock solution of 2 mg/mL. The stock solutions were successively diluted in the mix of methanol; acetonitrile 2:1 was used to obtain the series of working solutions at different concentration levels for external calibration. At the same time, volumes of 0.3 mL QC deproteinated samples were spiked with different volumes of standard solutions.

According to the “Guidance for Industry-Bioanalytical Method Validation” recommended by the US Food and Drug Administration, the UHPLC-QTOF-ESI^+^-MS method was validated to evaluate the linearity, specificity, precision, accuracy, limit of detection (LOD) and limit of quantification (LOQ). Two calibration curves were generated: an external standard calibration curve which was (1) made by diluting standard solutions in the mobile phase and an internal standard curve, and (2) whose linearity was determined for QC samples spiked with different volumes of each standard solution. The mean peak area of three replicate measurements at each concentration was calculated.

The LOD was the lowest concentration of analyte in the test sample that can be reliably distinguished from zero to signal/noise ratio ≥ 10. The LOQ was the lowest concentration of analyte that can be determined with an acceptable repeatability and trueness (signal/noise ratio ≥ 10 and SD values ≤ 40%).

## 3. Results

### 3.1. Untargeted Multivariate and Univariate Analyses

The untargeted multivariate and univariate UHPLC-QTOF-ESI^+^-MS analyses recorded and presented previously [27] provided seven gut-derived molecules, from serum and urine, that expressed strong biomarker potential.

Taking into consideration the separation data accumulated from UHPLC-QTOF-ESI^+^-MS analysis, the matrices representing Retention Times (RT), the mass-to-charge ratio (*m*/*z*) and the Area Under Curve (AUC) values corresponding to the seven targeted metabolites were used for the subsequent statistical analysis, as presented in Table 2. In addition, Table 2 displays the mean peak intensities (PI) of DKD group and of C group, and the ratio of mean DKD/mean C group.

According to the data recorded, serum arginine and hippuric acid may discriminate with significant *p*-values and higher AUC values in the group P vs. group C. These two metabolites had decreased while the other metabolites had increased values in the P group in serum.

Similarly, urinary p-cresyl sulfate, arginine, L-acetylcarnitine and hippuric acid discriminate P vs. C group based on p-values and AUC values. At the same time, indoxyl sulfate and buteonylcarnitine had increased, while the other metabolites had decreased values in the P group, in urine.

The untargeted univariate analysis, based on one-way ANOVA and Fisher’s LSD algorithms, permitted the discrimination of the seven metabolites, between subgroups (P1 vs. P2, P 3, C).

In brief, based on the statistical methods mentioned above, the following metabolites considered for the targeted analysis were arginine, hippuric acid, sorbitol, L-acetylcarnitine, indoxyl suflate, butenoylcarnitine—in serum; and arginine, hippuric acid, p-cresyl sulfate, L-acetyl carnitine, indoxyl suflate and buteonylcarnitine—in urine. Their discrimination between subgroups based on one-way ANOVA and Fisher LSD’s algorithms and their differences based on MS peak intensities are graphically displayed in Appendix A.

### 3.2. Targeted Analysis of Selected Metabolites

#### 3.2.1. Calibrations and Validation Parameters

The linear ranges [calibration curves and equations including the correlation coefficients (R^2^ values)], the Limit of Detection (LOD) and Limit of Quantification (LOQ) of each standard are given in Table 3. The correlation coefficients (R^2^) were higher than 0.9 for all standards in their linear range, showing a good linear relationship within linear ranges. All the LOD values were in the range 0.3–4 μM, and LOQ values were in the range 0.9–5.5 μM.

The validation of the LC-MS method for the quantitative evaluation of metabolites was performed using controlled additions of internal standard (DOXO) and each of the eight pure standards to QC extracts.

To the same volume of QC extracts (0.3 mL) 0.2 mL was added from each of the eight standard solutions (50 μM creatinine, 5 μM Arginine, Dimethylarginine, hippuric acid, L-Acetylcarnitine, 12.5 μM indoxyl sulfate and p-cresylsulfate, 7.5 μM sorbitol) as well as 3.4 μM of internal standard DOXO. Table 4 displays the initial concentrations of metabolites after mixing with the QC extract and the measured concentrations after the LC-MS analysis. The recovery percentage was calculated as a measure of the method reproducibility.

#### 3.2.2. Quantitative Evaluation and Statistical Analysis

The quantitative evaluation in serum and urine, based on the curve equations for each biomarker and for each group (C, P1, P2, P3), is presented in Table 5. The concentration of dimethylarginine is calculated according to the calibration with ADMA and represents the sum of asymmetric and symmetric dimethylarginine, since the LC-MS separation could not discriminate the two stereoisomers. Thus, dimethylarginine is expressed in ADMA units. The calibration curves are presented in Appendix A and the comparison between untargeted and targeted analyses of the metabolites expressed in mean PI DKD/mean PI C ratios are given in Appendix A.

The statistical analysis was performed by (1) one-way ANOVA with Bonferroni correction analysis, a Chi squared test and a Kruskal–Wallis test, which allowed for the differentiation of clinico-biological features between subgroups as presented in Table 1 and (2) a Mann–Whitney test which permitted the evaluation of metabolite differences between subgroups; the data are presented in Table 5.

Subsequently, the data obtained were correlated with references from the HMDB for each metabolite as can be seen in Table 6.

## 4. Discussion

The present study represents a second step taken into understanding the complex dynamics of gut-derived metabolites in different stages of DKD, with special attention on the normoalbuminuric subgroup (P1). It comes as an extension to a previous untargeted metabolite assessment of serum and urine of T2DM suffering with DKD [27]. The subsequent analysis of the same samples was performed by UHPLC-QTOF-ESI+-MS-targeted techniques, and the metabolites that expressed strong biomarker potential in our previous study, were selected. The incriminated metabolites are such as the following: arginine, dimethyl arginine, hippuric acid, indoxyl sulfate, p-cresyl sulfate, L-acetylcarnitine, buteonylcarnitine and sorbitol.

### 4.1. Free Amino-Acid Arginine and Its Metabolite, ADMA—Their Involvement in DKD

Our results successively reflect decreasing concentrations (μΜ) of arginine, in serum, from the control group to normo- to macroalbuminuria. In contrast, ADMA has a slightly increased trend in the serum when comparing the subgroups. Subsequent to the statistical analysis applied, it was observed that serum arginine displays differences when comparing C group with P subgroups and P1 with P2, while ADMA shows good differentiation between C and P subgroups. In urine, it can be observed that arginine and ADMA have a simultaneously increasing trend, excepting P1 vs. P2 where the levels of both metabolites decrease.

These results are consistent with previous studies that had also described low levels of arginine and high levels of ADMA in the serum of patients suffering from CKD [11]. By comparing these results with concentrations provided by the HMDB we may acknowledge that arginine ranges from 81.4 +/− 19.3 uM in serum [29], and 2.37 μM/mmol creatinine, in urine [30] when adjusted for age and sex and in normal conditions. Going further, Jayachandran et al. revealed a cut-off point of 0.66 μΜ for ADMA in T2DM without CKD, while Davoudi et al. found a cut-off point of 0.39 μΜ for asymmetric dimethylarginine in normoalbuminuric DKD [12,31].

Arginine is an exogenous, non-essential amino acid and represents a precursor of a large number of metabolites, being involved in multiple chemical reactions into the digestive tract [32,33]. By the action of nitric oxide synthetase (NOS), arginine converts to nitric oxide (NO). On the other hand, its cleavage by protein arginine methyl transferases (PRMTs) results in the formation of methylated arginines (ADMA and symmetric dimethylarginine). ADMA inhibits directly the NO formation through NOS blockage and its dynamic is of great interest in DKD because of its renal excretion [34]. With regard to DKD evolution, Jayachandran et al. performed a targeted analysis on ADMA and they observed that its levels were increasing even in the T2DM group without DKD [12]. Furthermore, several years later, they performed a study on rat kidney isolated cells and discovered that in a hyperglycemic state, ADMA metabolism is dysregulated and is implicated in the pathological mechanisms of renal fibrosis [35].

By integrating our results with data found in literature, we may postulate that the underlying mechanism by which arginine and ADMA interfere with DKD pathogenesis relies on the fact that glomerular endothelial cells, in conditions of hyperglycemia, become saturated with glucose. As a result of this, the glycolysis normal process starts taking deleterious routes, with subsequent ROS production and enhanced oxidative stress [36]. Arginine is known to be an antioxidant factor, and by being downregulated in normoalbuminuric patients, it strengthens the fact that a pro-oxidant state is under way [37]. This exerts toxic effects on endothelial cells and glycocalyx, with subsequent enlargement of the fenestrations which permits the passage of albumin and proinflammatory factors. Oxidative stress also determines an endothelial–to–mesenchymal cell transition, triggering incipient fibrotic processes [38]. The facts that arginine levels are low and ADMA levels are high even in the P1 group and they display differences between subgroups, points to their subtle implication in the pathogenesis of early DKD by maintaining (1) defects in the vasodilatation of renal microvasculature; (2) impairment in the normal glycolysis process; and (3) oxidative stress with subsequent ROS production.

### 4.2. Uremic Toxins (Hippuric Acid, Indoxyl Sulfate and p-Cresyl Sulfate) May Be Involved in Incipient DKD

Our study points out decreasing serum concentrations of hippuric acid (C < P1 < P2 < P3). In urine, its concentrations successively increase from C to P1–P3 subgroups. These findings are in agreement with previous studies which describe lower serum levels of hippuric acid [16] and higher urinary levels of this metabolite [18], in the context of hyperglycemia when compared to controls. In contrast, other authors have found lower urinary levels of hippuric acid in diabetic patients vs. controls [16,17,39]. When compared to the HMDB, hippuric acid ranges from 5–20 mM in serum and 200+/− mM/mM creatinine, in urine as described in Table 6. According to the statistical comparison between subgroups, the results presented in Table 5 reflect that hippurate may only be considered as an early DKD biomarker in serum. In urine, it does not discriminate C from DKD subgroups.

Renal proximal tubular cells encompass the highest density of mitochondria within the kidney being the main site along with the liver of hippuric acid production [40]. The decline of hippuric acid concentration from C group to DKD subgroups (P1, P2, P3) may be explained by a defective mitochondrial function on proximal tubular epithelial cells. This process seems to develop early in the course of DKD and as this condition progresses. Furthermore, phenylalanine, a precursor of hippuric acid, is present at low levels according to our previous study [27], and in consequence the phenylalanine pathway is impaired, with subsequent low concentrations of hippuric acid in serum. The fact that these modifications in hippuric acid concentrations happen in the P1 group and between subgroups suggests its possible involvement in the early stages of DKD as a diagnosis biomarker. However, these statements need to be warranted by correlations with specific renal tubular markers in further studies.

Regarding indoxyl sulfate, an increment of its serum concentrations was observed from C to normo- and macroalbuminuria, following the same pattern in urine. By subsequent analysis, reflected in Table 5, we have observed that indoxyl sulfate expresses a good biomarker potential in serum.

Indoxyl sulfate is a well-known uremic toxin and a derivative of tryptophan metabolism, as described in our previous study [27]. Other studies have highlighted the dynamic of indoxyl sulfate in DKD. Ng et al. performed a urine metabolomic analysis on DKD patients with low GFR and normoalbuminuria and discovered high levels of indoxyl sulfate in this category [19]. On the other hand, Niewczas et al. determined indoxyl sulfate in serum, with levels that ranged from 8 +/− 9.2 μM/L in patients with T2DM, and demonstrated that this metabolite is not a factor of progression to ESRD [20].

Indoxyl sulfate seems to be implicated in multiple reactions concerning the glomerulus and proximal tubule. For example, high levels of indoxyl sulfate represent a source from which emerges a cascade of unfavorable events regarding glomerular microvascularization through activation of aryl hydrocarbon receptor (AhR) [41]. It is a promoter of inflammation, vascular calcification and microtrombi formation, resulting in endothelial cell apoptosis and an enhanced permeability of glomerular filtration membrane. On a tubular level, indoxyl sulfate acts on its specific receptors on tubular epithelial cells, which are organic anion transporters (OATs) and triggers pro-inflammatory processes that culminate with cell cycle arrest [42]. According to previous studies and our results, indoxyl sulfate may be a candidate biomarker of incipient DKD in serum. Its presence in high quantity may be a mirror of incipient glomerular endothelial dysfunction and tubular epithelial cell apoptosis in the diabetic model.

Additionally, p-cresyl sulfate was quantified in urine, where we have observed an increasing trend, with its concentrations being highly discriminative when comparing the C group with the other subgroups and P1 vs. P2, expressing significant *p*-values. When confronting these results with HMDB, p-cresyl sulfate ranges from 0.3 to 5.5 μM in urine in normal state.

Tyrosine, along with phenylalanine, are compounds originating from diet and represent the main source of phenols produced into the digestive tract. Tyrosine, in particular, generates phenol and p-cresol through bacterial fermentation. Thus, the formation of p-cresyl sulfate derives from the hepatic metabolization of p-cresol. P-cresyl sulfate elimination occurs through glomerular filtration and tubular secretion [43]. Poveda et al. demonstrated in vitro that p-cresyl sulfate is directly involved in renal proximal tubular epithelial cell damage, by activating inflammatory processes and subsequent cell apoptosis [44]. Koppe et al. described the implication of p-cresyl sulfate in the processes of insulin resistance in patients with CKD [45]. Although several authors have described p-cresyl sulfate concentration in serum [46,47], few studies have been developed with regard to urinary concentrations of p-cresyl sulfate.

Our study suggests an early involvement of p-cresyl sulfate in the mechanisms of incipient DKD. This metabolite is considered to follow the same pathogenic route as indoxyl sulfate by producing endothelial and proximal tubular damage, as a consequence of a pro-oxidant environment. Higher levels of p-cresyl sulfate in our normoalbuminuric group in urine suggest that its secretion is not dependent on albuminuria and may determine tubular proximal damage in vivo as well.

### 4.3. Acylcarnitines (L-acetylcarnitine and Buteonyl Carnitine) Dynamic in Normoalbuminuria DKD

With regard to L-acetylcarnitine, our study pointed out that its concentrations in serum have a maximum peak in the normoalbuminuric subgroup (P1) when compared with the other subgroups (C, P2, P3). In urine, its concentrations have a tendency to increase progressively, from C group to macroalbuminuria. According to HMDB, the levels of L-acetylcarnitine in serum and urine range from 5–7 μM and 1–3 μM, respectively. By performing additional analyses, we have observed that L-acetylcarnitine cannot be considered a biomarker of DKD, since it did not reach the statistical significance in the comparison between subgroups.

Butenoylcarnitine is another short-chain carnitine which has an initial decrease in serum, from C to P1 group, and a later increase from P2 and P3 subgroups, respectively. In urine, its trend reveals decreased levels in control group and equal levels between DKD subgroups. According to Table 5, butenoylcarnitine expresses good biomarker potential in serum.

Fatty acids in large amounts exceed the ability of mitochondria to process them. Consequently, the accumulation of short-chain acylcarnitines in serum and urine, such as butenoylcarnitine, reflect mitochondrial dysfunction. The levels of this metabolite, in serum, were found to be elevated in patients with DKD by Pena et al. [48,49]. In our study, buteonylcarnitine was found to be discriminative between P1 vs. P2 subgroups, in serum. Its high excretion in the normoalbuminuric group, besides mitochondrial dysfunction, may also point to a loss of tubular epithelial cells which have a high density of mitochondria, with subsequent impairment in its renal tubular reabsorption.

### 4.4. Sorbitol

In our study, sorbitol concentrations were observed to be lower in the P1 group when compared with C group. Going further, the levels of sorbitol tend to increase as albuminuria develops, reaching high levels in the macroalbuminuric group. It also shows a significant distinction between C, P1 and P2. In a diabetic state, large amounts of glucose determine polyol pathway activation, with subsequent formation of sorbitol. The enzymes used as substrate for the production of sorbitol are most aggregated in organs vulnerable to develop complications due to diabetes [25,50]. Levels of NADPH and NAD+ are dysregulated in a context of high sorbitol production, with this phenomenon interfering with NO production [26]. Thus, sorbitol may express biomarker potential of early DKD through the activation of the polyol pathway.

The strengths of our study reside in the fact that there is a lack of targeted metabolomic studies regarding early DKD with a focus on gut-derived metabolites. The cut-off points of these metabolites are not well established for certain diseases; thus, this study may contribute to their future standardization. Therefore, our paper brings a novelty designated to deepen the understanding of metabolic disturbances that occur in the very early phases of DKD.

Nevertheless, we may acknowledge that our study has some limitations due to a low number of participants, being a pilot and a cross-sectional study. We cannot exclude the interference of lifestyle habits, which are known to modulate metabolite fluctuations due to dietary intake of certain precursors. Furthermore, by the lack of stool sample analysis, our study can only point out the gut provenance of the metabolites according to literature data. Interindividual fluctuations of serum levels of glucose cannot be excluded even though patients with uncontrolled diabetes were not enrolled. In addition, the comparison with HMDB is only informative as age, race and sex differences are not uniform.

### 4.5. Transition from Biomarker Discovery to Clinical Practice

The aforementioned biomarker discovery may have multiple clinical implications. First, the determination of these molecules may provide biological panels for early detection of DKD before the occurrence of albuminuria or the decline of eGFR. Second, studies performed on L-arginine supplementation in patients with DKD or NDKD indicate an improvement on oxidative stress and subsequent albumin loss [37,51]. Third, the usage of probiotics in individuals with DKD is gaining more and more attention. Recent studies now point to the fact that probiotics may (1) alleviate and maintain the permeability of intestinal wall; (2) decrease the absorption of metabolites in systemic circulation, thus deviating the track of metabolite excretion; (3) reduce oxidative stress and inflammation in the renal endothelium; and (4) ameliorate renal function [52].

## 5. Conclusions

In summary, our study emphasizes the fact that arginine, dimethylarginine, hippuric acid, indoxyl sulfate, butenoylcarnitine and sorbitol may be considered as putative biomarkers involved in the early metabolic disturbances of DKD in serum. In addition, urine analysis reveals p-cresyl sulfate may be a candidate biomarker of incipient DKD in urine. Quantification of gut-derived metabolites provides novel insights regarding amino acids, uremic toxins and acylcarnitines which may later contribute to their standardization and usage in metabolic individualized biological panels with regard to early DKD diagnosis.

## Figures and Tables

**Table 1 biomolecules-13-01086-t001:** Clinical and biological characteristics of the participants.

	C	P1	P2	P3	*p*-Value
Subject enrolled (nr.)	20	30	30	30	0.601 *
**Clinical characteristics**					
Female (nr.,%)	8 (40%)	16 (53.34%)	17 (56.67%)	15 (50%)	0.393 **
DM duration (y)	0	9.6 ± 3.99	9.7 ± 3.99	12.78 ± 3.35	<0.001 ***
Diabetic polineuropathy (nr, %)	0	5 (16%)	9 (30%)	17 (56.6)	<0.001 **
Diabetic retinopathy (nr.,%)	0	6 (20%)	12 (40%)	20 (66.6%)	<0.001 **
**Biological parameters**					
UACR (mg/g)	5 ± 0.23	7.38 ± 3.22	45.42 ± 57.08	319.86 ± 585.80	<0.001 ***
HbA1c (%)	4.98 ± 0.23	5 ± 0.23	6.42 ± 1.29	7.15 ± 1.60	<0.001 *

DM—diabetes mellitus; HbA1c—hemoglobin A1c; UACR—urinary albumin/creatinine ratio; data reported as means ± standard deviation; the comparison between subgroups was based on * one-way ANOVA with Bonferroni correction, ** Chi squared test, *** Kruskal-Wallis test, analyses which provided the *p*-values.

**Table 2 biomolecules-13-01086-t002:** The classification of metabolites that discriminate the group DKD from group C, based on mean PI in DKD group, mean PI in C group, ratio DKD/C, RT, *m*/*z* and AUC values in serum and urine.

Serum									
*m*/*z*	Identification	PI DKD Group		PI C Group		Ratio DKD/C		RT (min)	AUC
		Mean	±SD	Mean	±SD				
175.1306	Arginine	100,039.21	47,525.72	124,331.91	25,243.14	0.80		1	0.5
180.1716	Hippuric acid	111,747.94	46,825.82	123,619.30	11,537.92	0.90		11	0.7
183.0940	Sorbitol	98,996.89	37,792.49	99,647.14	16,840.26	0.99		1	0.5
204.1369	L-Acetylcarnitine	201,181.48	108,608.92	199,328.56	74,900.62	1.01		1	0.6
214.2676	Indoxyl sulfate	48,652.85	17,782.53	40,115.19	2429.85	1.21		12	0.6
230.2668	Butenoyl carnitine	87,016.09	34,019.37	80,790.60	170.13	1.08		11	0.6
**Urine**									
***m*/*z***	**Identification**	**PI DKD Group**		**PI C Group**		**Ratio DKD/C**	**RT (min)**		**AUC**
		**Mean**	**±SD**	**Mean**	**±SD**				
175.1306	Arginine	14,288.87	7608.37	13,479.29	4687.36	1.06	1		0.7
180.1716	Hippuric acid	294,865.25	229,329.22	267,442.47	149,277.77	1.10	11		-
204.1369	L-Acetylcarnitine	16,892.46	13,440.22	7514.83	1256.62	2.25	1		0.5
214.2676	Indoxyl sulfate	16,693.72	9049.71	5529.51	1777.29	3.02	12		1
189.1594	p-Cresylsulfate	16,814.83	13,305.89	8798.191	4015.74	1.91	1		0.8
230.2668	Butenoyl carnitine	17,342.30	9668.83	4955.82	170.13	3.50	11		1

Data presented as means ± SD; AUC—area under curve; *m*/*z*—mass to charge ratio; PI—peak intensity; RT—retention time.

**Table 3 biomolecules-13-01086-t003:** Validation parameters (linear range, curve equation, R^2^, LOD and LOQ) for each of the seven molecules selected as potential biomarkers.

Name	Linear Range (μM)	Curve Equation	R^2^	LOD (μM)	LOQ (μM)
Arginine	2–40	y = 2476.6x + 427.61	0.999	0.2	0.8
Hippuric acid	0.5–10	y = 5097.7x − 441.68	0.999	0.2	0.8
p-Cresylsulfate	2.5–40	y = 2717.3x − 413.46	0.999	0.2	0.8
L-Acetylcarnitine	1–5	y = 36813x − 3879.2	0.994	0.2	1.0
Indoxyl sulfate	0.5–25	y = 8157x − 1240.8	0.999	0.2	0.8
Sorbitol	0.2–4	y = 39811x − 1472.8	0.999	0.15	0.8

**Table 4 biomolecules-13-01086-t004:** The recovery percentage (%) calculation from the measured concentrations of internal standard (IS) and each metabolite (pure standard) comparative to their initial concentrations, after addition to the QC extract.

Urine Metabolite	Initial Concentration (μM)	Measured Concentration (μM)	Recovery (%)
Arginine	2	1.8	91
Hippuric acid	2	1.9	94
p-Cresylsulfate	5	4.5	90
L-Acetylcarnitine	2	1.8	93
Indoxyl sulfate	5	4.8	96
Sorbitol	3	2.6	87
IS (DOXO)	1.4	1.3	89

**Table 5 biomolecules-13-01086-t005:** The mean values and standard deviations (±SD) of serum (μM) and urine concentrations (μM/μM creatinine) of the potential biomarkers targeted in this study for groups C and subgroups P1, P2 and P3.

Blood Serum (μM)	Control (*n* = 20)	P1 (*n* = 30)	P2 (*n* = 30)	P3 (*n* = 30)
Arginine	50 (10) ^†,^*	44 (10.2) ^▲^	39 (6.6)	38 (7.8)
Dimethyl Arginine	0.9 (0.2) ^†,^*	1.1 (0.3)	1.1 (0.5)	1.2 (0.4)
Hippuric acid	24 (2.4) ^†,⁑^	22.7 (1.2)	22.4 (1.7)	21 (5)
Indoxyl sulfate	5 (0.5) *	5.1 (0.5) ^▲^	6.6 (5.2)	6.6 (0.6) ^♦^
L-Acetylcarnitine (AC)	5.5 (2.1)	5.7 (2.1)	5.4 (1.7)	5.6 (1.7)
Butenoylcarnitine (eq AC)	2.3 (0.1) *	2.3 (0.1) ^♣^	2.6 (0.4)	2.5 (0.5)
Sorbitol	2.5 (0.5) ^†^	2.3 (0.1) ^♣^	2.7 (0.3)	2.6 (0.4)
**Urine (μM/μM creatinine)**	**Control**	**P1**	**P2**	**P3**
Arginine	5.3 (1.7)	6.1 (2.9) ^▲^	5 (3.4)	5.7 (2.7)
Dimethyl Arginine	3.1 (0.4) ^⁑^	35.6 (7.5)	21 (4.6)	50.5 (11.1)
Hippuric acid	52.6 (29.4)	54.7 (24.3)	59.5 (73.3)	59.5 (38.3)
Indoxyl sulfate	0.8 (0.4) ^♦,^*	2.1 (1.1)	2.1 (1.5)	2.5 (1.2)
p-cresyl sulfate	3.4 (1.6) ^†,⁑^	5.7 (5.1) ^▲^	5.9 (3.5)	7.4 (6.4)
L-Acetylcarnitine (AC)	0.3 (0.1)	0.4 (0.3)	0.4 (0.3)	0.6 (0.5)
Butenoylcarnitine (eq AC)	0.2 (0.1) ^♦,^*	0.5 (0.2)	0.5 (0.3)	0.5 (0.3)

Metabolites concentrations are presented as medians and standard deviations. Statistical significance between healthy controls and normoalbuminuric group, ^♦^
*p* < 0.001; ^†^
*p* > 0.001 and *p* < 0.05; Statistical significance between normoalbuminuric group and microalbuminuric group, ^♣^
*p* < 0.001; ^▲^
*p* > 0.001 and *p* < 0.05; Statistical significance between microalbuminuric group and macroalbuminuric group, ^♦^
*p* < 0.001; Statistical significance between healthy controls vs. normoalbuminuric group vs. microalbuminuric group vs. macroalbuminuric group; * *p* < 0.001, ⁑ *p* > 0.001 and *p* < 0.05; *p*-values based on Mann–Whitney test.

**Table 6 biomolecules-13-01086-t006:** Normal ranges for the targeted metabolites based on the HMDB.

Metabolite	Normal Range Based on HMDB		Link
	Serum (μM)	Urine (μM/μM Creatinine)	
ADMA	0.3–1	1–12	https://hmdb.ca/metabolites/HMDB0001539 accessed on 29 April 2023
Hippuric acid	5–20	200+/−	https://hmdb.ca/metabolites/HMDB0000714 accessed on 29 April 2023
Indoxyl sulfate	1–4	10–20	https://hmdb.ca/metabolites/HMDB0000682 accessed on 29 April 2023
p-Cresyl sulfate	Data not found	1.3 (0.3–5.5)	https://hmdb.ca/metabolites/HMDB0011635 accessed on 29 April 2023
L-Acetylcarnitine	5–7	1–3	https://hmdb.ca/metabolites/HMDB0000201 accessed on 29 April 2023
Butenoylcarnitine	Data not found	Data not found	https://hmdb.ca/metabolites/HMDB0249460 accessed on 29 April 2023
Sorbitol	1–3	-	https://hmdb.ca/metabolites/HMDB0000247 accessed on 29 April 2023

## Data Availability

The data that support the findings of this study are available from the corresponding author upon reasonable request.

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
