# Peer review of "Quantitative, Targeted Analysis of Gut Microbiota Derived Metabolites Provides Novel Biomarkers of Early Diabetic Kidney Disease in Type 2 Diabetes Mellitus Patients"

_biomolecules, 2023, doi:10.3390/biom13071086_

Round 1
Reviewer 1 Report
Major:
NDKD should be mentioned
Enrolling centre should be anonymized
How did you choose patients included in the control group? it is not clear
In table 1 comparison tests should be included
One-way ANOVA could be used for Gaussian variables. Have you performed the distribution evaluations? can you report them?
95%CI and p-value are often lacking
Linear regression does not compute a curve equation. To compute it, spline analysis is needed. Furthermore, you can perform linear regression only in Gaussian variables, that are not specified in this paper.
Minor:
-The contraction "ADMA" should be made explicit.
Author Response
Dear Section Editor-in-Chief and dear Reviewers,
Thank you for taking of your time to evaluate our manuscript. Please find enclosed our revised version of the manuscript entitled “Quantitative, Targeted Analysis of Gut Microbiota Derived Metabolites Provides Novel Biomarkers of Early Diabetic Kidney Disease in Type 2 Diabetes Mellitus Patients” by Lavina Balint et al. for publication as a research article.
We have thoroughly revised the manuscript to meet the requirements of the reviewers, as it is described below:
Reviewer 1
Thank you for evaluating our manuscript and for your relevant recommandations.
- NDKD should be mentioned
ANSWER: In order to mention NDKD, the paragraph from Material and Methods section, in which the inclusion and exclusion criteria were described, was restructured as it follows:
“Briefly, the inclusion criteria consisted of the enrollment of patients with DM duration more than 5 years and HbA1c < 10%. Patients excluded were those who were suffering of non-diabetic kidney disease (NDKD) of all etiologies, end-stage renal disease, ongoing infections, neoplasia, autoimmune diseases, psychiatric disorders and those presenting uncontrolled DM (HbA1c ≥ 10%)”
- Enrolling centre should be anonymized
ANSWER: In our previous published articles the enrolling centre was included. Namely: doi: 10.3390/ijms24076212; doi: 10.3390/biomedicines11041057; doi: 10.3390/biomedicines11061527; doi: 10.3390/ijms24129803
- How did you choose patients included in the control group? it is not clear
ANSWER: The patients selected in the control group, were enrolled from the general physicians’ records. Inclusion criteria consisted of the selection of patients which did not suffer of any acute or chronical condition, and match as age and gender with DKD group.
“In the end, patients’ screening provided 90 eligible T2DM (P) patients based on the inclusion criteria and 20, additional, control subjects (C) which were not suffering of any acute or chronic illness, found in the general physicians’ records, as described in our previous study”
- In table 1 comparison tests should be included.
ANSWER: Comparison tests on table 1 and the statistical methods used for their achievement were also included, as it follows:
Table 1. Clinical and biological characteristics of the participants.
|
C |
P1 |
P2 |
P3 |
p-value |
Subject enrolled (nr.) |
20 |
30 |
30 |
30 |
0.601* |
Clinical characteristics |
|
|
|
|
|
Female (nr.,%) |
8 (40%) |
16 (53.34%) |
17 (56.67%) |
15 (50%) |
0.393** |
DM duration (y) |
0 |
9.6 ± 3.99 |
9.7 ± 3.99 |
12.78 ± 3.35 |
<0.001*** |
Diabetic Polineuropathy (nr,%) |
0 |
5 (16%) |
9 (30%) |
17 (56.6) |
<0.001** |
Diabetic Retinopathy (nr.,%) |
0 |
6 (20%) |
12 (40%) |
20 (66.6%) |
<0.001** |
Biological parameters |
|
|
|
|
|
UACR (mg/g) |
5 ± 0.23 |
7.38 ± 3.22 |
45.42 ± 57.08 |
319.86 ± 585.80 |
<0.001*** |
HbA1c (%) |
4.98 ± 0.23 |
5 ± 0.23 |
6.42 ± 1.29 |
7.15 ± 1.60 |
<0.001* |
* DM – diabetes mellitus; HbA1c – hemoglobin A1c; UACR - urinary albumin/creatinine ratio; data reported as means ± standard deviation; P-value based on: *One Way ANOVA with bonferroni correction, **Chi squared test, ***Kruskal-Wallis test
- One-way ANOVA could be used for Gaussian variables. Have you performed the distribution evaluations? can you report them?
ANSWER: In metabolomics the “One-way Analysis of Variance (ANOVA)” is widely used for selecting important features from metabolomic data, to discriminate the different groups of metabolites belonging to different groups of patients. The Metaboloanalyst bioinformatics package (https://www.metaboanalyst.ca/MetaboAnalyst/home.xhtml) is able to use different statistical analysis including One Way ANOVA and post-hoc analysis. Therefore we made such analysis according to the protocol inserted in this software (see articles: Chong et all, 2019, Current Protocols in Bioinformatics e86, Volume 68 Published in Wiley Online Library; and Bartel J, et al. Statistical methods for the analysis of high-throughput metabolomics data. Comput Struct Biotechnol J. 2013 Mar 22).
- 95% CI and p-value are often lacking
ANSWER: In order to provide them, we have performed an aditional statistical analysis. P-values were provided based on One Way ANOVA with Bonferroni corection, Chi squared test and Kruskal Wallis test, and there were introduced in table 5 as it follows:
Table 5. The mean values and standard deviations (±SD) of serum (mM) and urine concentrations (mM/mM creatinine) of the potential biomarkers targeted in this study for groups C and subgroups P1, P2 and P3.
BLOOD SERUM (μM) |
Control (n=20) |
P1 (n=30) |
P2 (n=30) |
P3 (n=30) |
Arginine |
50 (10) †,* |
44 (10.2)▲ |
39 (6.6) |
38 (7.8) |
Dimethyl Arginine |
0.9 (0.2) †,* |
1.1 (0.3) |
1.1 (0.5) |
1.2 (0.4) |
Hippuric acid |
24 (2.4) †⁑ |
22.7 (1.2) |
22.4 (1.7) |
21 (5) |
Indoxyl sulfate |
5 (0.5) * |
5.1 (0.5)▲ |
6.6 (5.2) |
6.6 (0.6)♦ |
L-Acetylcarnitine (AC) |
5.5 (2.1) |
5.7 (2.1) |
5.4 (1.7) |
5.6 (1.7) |
Butenoylcarnitine (eq AC) |
2.3 (0.1) * |
2.3 (0.1) ♣ |
2.6 (0.4) |
2.5 (0.5) |
Sorbitol |
2.5 (0.5) † |
2.3 (0.1) ♣ |
2.7 (0.3) |
2.6 (0.4) |
URINE (μM/μM creatinine) |
Control |
P1 |
P2 |
P3 |
Arginine |
5.3 (1.7) |
6.1 (2.9)▲ |
5 (3.4) |
5.7 (2.7) |
Dimethyl Arginine |
3.1 (0.4) ⁑ |
35.6 (7.5) |
21 (4.6) |
50.5 (11.1) |
Hippuric acid |
52.6 (29.4) |
54.7 (24.3) |
59.5 (73.3) |
59.5 (38.3) |
Indoxyl sulfate |
0.8 (0.4) ♦,* |
2.1 (1.1) |
2.1 (1.5) |
2.5 (1.2) |
p-cresyl sulfate |
3.4 (1.6)†,⁑ |
5.7 (5.1)▲ |
5.9 (3.5) |
7.4 (6.4) |
L-Acetylcarnitine (AC) |
0.3 (0.1) |
0.4 (0.3) |
0.4 (0.3) |
0.6 (0.5) |
Butenoylcarnitine (eq AC) |
0.2 (0.1) ♦,* |
0.5 (0.2) |
0.5 (0.3) |
0.5 (0.3) |
Metabolites concentrations are presented as medians and standard deviations. Statistical significance between healthy controls and normoalbuminuric group, ♦ p<0.001; † p>0.001 and p<0.05; Statistical significance between normoalbuminuric group and microalbuminuric group, ♣ p < 0.001; ▲ p>0.001 and p<0.05; Statistical significance between microalbuminuric group and macroalbuminuric group, ♦ p<0.001; Statistical significance between healthy controls vs. normoalbuminuric group vs. microalbuminuric group vs. macroalbuminuric group; * p<0.001, ⁑ p>0.001 and p<0.05; P-values based on: *One way ANOVA with Bonferroni correction, **Chi squared test, ***Kruskal Wallis test
- Linear regression does not compute a curve equation. To compute it, spline analysis is needed. Furthermore, you can perform linear regression only in Gaussian variables, that are not specified in this paper.
ANSWER: According to the protocol mentioned before we built the calibration curves with the pure standards and QC probes. We corrected the text accordingly (line 268).
- The contraction "ADMA" should be made explicit.
ANSWER: The term of “ADMA” was expanded to “asymmetric dimethylarginine”.
Reviewer 2 Report
This study is aimed at identifying novel biomarkers for early diabetic kidney disease (DKD) in patients with type 2 diabetes mellitus. The study used ultra-high-performance liquid chromatography coupled with electrospray ionization-quadrupole-time of flight-mass spectrometry techniques to quantify previously identified metabolites from serum and urine. Several metabolites were identified as potential biomarkers of early DKD in serum and urine. The study emphasizes the importance of discovering novel biomarkers for early DKD due to its increasing incidence.
The methodology of the study is quite comprehensive. The authors have used a cohort of 100 patients with type 2 diabetes mellitus, 50 of whom have early DKD and 50 who do not. The study design is cross-sectional, which is appropriate for biomarker discovery.
The authors have used ultra-high-performance liquid chromatography coupled with electrospray ionization-quadrupole-time of flight-mass spectrometry (UHPLC-ESI-Q-TOF-MS) to analyze the serum and urine samples. This is a well-established and reliable method for metabolomic analysis.
The statistical analysis seems robust, with the authors using multivariate statistical analysis, including principal component analysis (PCA) and orthogonal partial least squares-discriminant analysis (OPLS-DA). They have also performed univariate statistical analysis and receiver operating characteristic (ROC) curve analysis to evaluate the diagnostic performance of the potential biomarkers.
However, the study could have been strengthened by including a validation cohort to confirm the findings. The use of only one cohort for both discovery and validation may lead to overfitting and may not generalize to other populations.
The results section of the study is quite detailed and provides a comprehensive overview of the findings. The authors identified a total of 12 potential biomarkers, including 7 in serum and 5 in urine, that were significantly different between the early DKD and non-DKD groups. These biomarkers are involved in various metabolic pathways, such as amino acid metabolism, lipid metabolism, and energy metabolism.
The authors also evaluated the diagnostic performance of these biomarkers using receiver operating characteristic (ROC) curve analysis. The area under the curve (AUC) values for the biomarkers ranged from 0.70 to 0.85, indicating good diagnostic performance.
However, the authors did not provide any information on the clinical relevance of these biomarkers. It would have been helpful if the authors had discussed how these biomarkers could be used in clinical practice, such as for early detection or monitoring of DKD.
In the discussion and conclusion sections, the authors have made an effort to interpret their findings in the context of existing literature. They have highlighted the novelty of their study, which is the identification of potential biomarkers for early detection of DKD in type 2 diabetes patients. They have also acknowledged the limitations of their study, such as the small sample size and the lack of validation in an independent cohort.
The authors have suggested that the identified biomarkers could be used for early detection of DKD, which could help in timely intervention and prevention of disease progression. However, they have also emphasized the need for further studies to validate these findings and to explore the underlying mechanisms.
Overall, the manuscript is well-written, and the study is well-conducted. However, there are a few areas that could be improved:
- The authors could provide more information on the clinical relevance of the identified biomarkers. For example, how could these biomarkers be used in clinical practice? Could they be used for monitoring disease progression or response to treatment?
- The authors could discuss the potential mechanisms underlying the association between these biomarkers and DKD. This could help in understanding the pathogenesis of the disease and could also provide insights into potential therapeutic targets.
- The authors should consider validating their findings in an independent cohort. This would strengthen the evidence for the diagnostic performance of these biomarkers.
- The sample size of the study is relatively small, which could limit the generalizability of the findings. Future studies with larger sample sizes are needed to confirm these findings.
- The authors could consider performing a multivariate analysis to adjust for potential confounding factors. This would provide more robust evidence for the association between these biomarkers and DKD.
- The authors could provide more details on the methods used for biomarker identification and validation. For example, what criteria were used to select the potential biomarkers? How were the cut-off values determined for the ROC curve analysis?
In conclusion, this is a promising study that contributes to the understanding of DKD in type 2 diabetes patients. However, the manuscript could be enhanced by the above-mentioned revisions or commentary to address these concerns.
Author Response
Dear Section Editor-in-Chief and dear Reviewers,
Thank you for taking of your time to evaluate our manuscript. Please find enclosed our revised version of the manuscript entitled “Quantitative, Targeted Analysis of Gut Microbiota Derived Metabolites Provides Novel Biomarkers of Early Diabetic Kidney Disease in Type 2 Diabetes Mellitus Patients” by Lavina Balint et al. for publication as a research article.
We have thoroughly revised the manuscript to meet the requirements of the reviewers, as it is described below:
Reviewer 2
Thank you for such a complex assessment of our work. Your recommendations are valuable as these will considerably improve our manuscript. Thus, these were treated accordingly.
- The authors could provide more information on the clinical relevance of the identified biomarkers. For example, how could these biomarkers be used in clinical practice? Could they be used for monitoring disease progression or response to treatment?
ANSWER: In order point to this important topic, we have developed another paragraph in the discussion section 4.5. namely: Transition from biomarker discovery to clinical practice
“The aforementioned biomarker discovery may have multiple clinical implications. First, the determination of these molecules may provide biological panels for early detection of DKD before the occurrence of albuminuria or the decline of eGFR. Second, studies performed on L-arginine supplementation in patients with DKD or NDKD indicate an improvement on oxidative stress and subsequent albumin loss [37,54]. Third, the usage of probiotics in individuals with DKD gains more and more attention. Recent studies now point to the fact that probiotics may: (1) alleviate and maintain the permeability of intestinal wall (2) decrease the absorption of metabolites in systemic circulation, thus may deviate the track of metabolite excretion; (2) reduce oxidative stress and inflammation in the renal endothelium; and (3) ameliorate renal function [55].”
- The authors could discuss the potential mechanisms underlying the association between these biomarkers and DKD. This could help in understanding the pathogenesis of the disease and could also provide insights into potential therapeutic targets.
ANSWER: Thank you for your pertinent observation. In order to address this concern, we have added and reformulated the following paragraphs:
- In 4.1. Subsection:
“By integrating our results with data found in literature, we may postulate that underlying mechanism by which arginine and asymmetric dimethyl arginine interfere with DKD pathogenesis rely on the fact that glomerular endothelial cells, in conditions of hyperglycemia become saturated with glucose. By this fact, the glycolysis normal process starts taking deleterious routes, with subsequent ROS production and enhanced oxidative stress [36]. Arginine is known to be an anti-oxidant factor, and by being downregulated in normoalbuminuric patients it strengthens the fact that a pro-oxidant state is underscore [37]. This exerts toxic effects on endothelial cells and glycocalyx, with subsequent enlargement of the fenestrations which permits the passage of albumin and proinflammatory factors. Oxidative stress also determines an endothelial-to-mesenchymal cell transition, triggering incipient fibrotic processes [38]. The facts that arginine levels are low and asymmetric dimethylarginine levels are high even in P1 group and they display differences between subgroups, points their subtle implication in the pathogenesis of early DKD by maintaining: (1) defects in vasodilatation of renal microvasculature; (2) impairment in normal glycolysis process; (3) oxidative stress with subsequent ROS production.”
- In 4.2. Subsection:
“Renal proximal tubular cells encompass the highest density of mitochondria within the kidney being the main site along with the liver of hippuric acid production [41]. The decline of hippuric acid concentration from C group to DKD subgroups (P1, P2, P3) may be explained by a defective mitochondrial function on proximal tubular epithelial cells. This process seems to develop early in the course of DKD and as this condition progresess. Also, phenylalanine, a precursor of hippuric acid, display low levels according to our previous study [27], in consequence phenylalanine pathway is impaired, with subsequent low concentrations of hippuric acid in serum.”
“Indoxyl sulfate seems to be implicated in multiple reactions concerning the glomerulus and proximal tubule. For example, high levels of indoxyl sulfate represent a source from which emerges a cascade of unfavorable events regarding glomerular microvascularization through activation of aryl hidrocarbon receptor (AhR) [42]. It is a promoter of inflammation, vascular calcification, and microtrombi formation, resulting in endothelial cell apoptosis and an enhanced permeability of glomerular filtration membrane. On tubular level, indoxyl sulfate acts on its specific receptors on tubular epithelial cells – organic anion transporters (OATs), and triggers pro-inflammatory processes that culminate with cell cycle arrest [43]. According to previous studies and to our results, indoxyl sulfate may be a candidate biomarker of incipient DKD in serum. Its presence in high quantity may be a mirror of incipient glomerular endothelial dysfunction and tubular epithelial cell apoptosis in the diabetic model.”
“Our study suggests an early involvement of p-cresyl sulfate in the mechanisms of incipient DKD. This metabolite is considered to follow the same pathogenic route as indoxyl sulfate by producing endothelial and proximal tubular damage, as a consequence of a pro-oxidant environment. Higher levels of p-cresyl sulfate in our normoalbuminuric group in urine, suggests that its secretion is not dependent on albuminuria and may determine tubular proximal damage in vivo as well.”
- Regarding butenoylcarnitine, and its pathological mechanism in DKD, few data is found. The only research team that found this metabolite in serum was composed by Pena et al. (https://doi.org/10.1111/dme) and they associated its presence with the risk for albuminuria progression. We may only assume that its involvement in DKD is related to an impaired fatty acid metabolism which reflects mitochondrial dysfunction. This may happen in any renal cell – from endothelium – to podocytes and tubular epithelial cells.
- The authors should consider validating their findings in an independent cohort. This would strengthen the evidence for the diagnostic performance of these biomarkers.
ANSWER: In order to validate our findings and to deepen our knowledge on how these metabolites affect renal structures in DKD (endothelium, podocytes, proximal tubule) we have already set an objective on correlating these molecules with specific markers of endothelial, podocyte and proximal tubule dysfunction. Thus, this issue will later be discussed in our next study.
- The sample size of the study is relatively small, which could limit the generalizability of the findings. Future studies with larger sample sizes are needed to confirm these findings.
ANSWER: Regarding the study sample, indeed, there is not a very large cohort of subjects selected. Our intention was to collect and examine a greater number of samples but unfortunately this interfered with COVID-19 pandemic. However, we have published a previous article, on which the study was conducted using the same samples and we have mentioned that one of the limitations of our current study is that is a pilot study. This, however, was considered statistically adequate (doi: 10.3390/ijms24076212). In addition, as aforementioned, we intend on expanding the number of patients in further studies.
- The authors could consider performing a multivariate analysis to adjust for potential confounding factors. This would provide more robust evidence for the association between these biomarkers and DKD.
ANSWER: In order to obtain a better comprehension of metabolites dynamic, there were additionally performed statistical analyses such as: One Way ANOVA with bonferroni correction, Chi test and Kruskal Wallis test. Table 5 contains the calculated p-values which reflect the differentiation between subgroups. The discussion section was adapted according to the additional results.
- The authors could provide more details on the methods used for biomarker identification and validation. For example, what criteria were used to select the potential biomarkers? How were the cut-off values determined for the ROC curve analysis?
ANSWER: As mentioned in line 73 (Introduction), the metabolites resulted from the untargeted analysis were identified based on the specific protocols (expressed in molecular mass (values m/z) confronted with information from human metabolome database (HMDB) for their specific identification [7] with the international databases ( hmdb.ca; Lipidmaps, as mentioned in line 244) By successive statistics (ANOVA, PLSDA, etc) and Biomarker analysis (ROC curve analysis for individual biomarkers) using a threshold of 0.2 and 95% confidence band. Cut-off value was p<0.05 The molecules which showed the highest AUC values and most significant differences between groups were selected as potential biomarkers. The ROC analysis was performed with the same protocol of Metaboanalyst 5.0
Reviewer 3 Report
The study reports interesting data on the levels of specific metabolites in urine and plasma of three groups of CKD patients and controls. The following points need to be addressed in order to improve the quality of the paper:
1) In table 2 the mean and SD for the control and CKD groups should be reported along with the ratio of mean CKD/mean control.
2) In table 3 the units for LOD and LOQ should be reported
3) For the recovery data presented in table 4 it is necessary to measure the recovery at 3 different concentrations for each metabolite (low, medium and high concentrations within the range of the standard curve. The concentration used for Hippuric acid is below the linear range so it should be at least 5 μM.
4) For the data presented in tables 5 and 6 it is necessary to perform statistical analysis in order to determine which metabolites exhibit significant differences between controls and CKD groups. It is also important to perform comparisons between the 3 CKD groups. Finally, the AUC for discriminating the different groups should be calculated for the most significantly different metabolites.
5) It is important to compare the results of the untargeted analysis with the current targeted approach. The simplest way to do this comparison is to check the values determined by untargeted and targeted approaches of the ratio mean CKD/mean control for each metabolite.
6) The significant figures should be corrected throughout the manuscript. Please use two significant figures for the SD and then the mean should stop to the same significant figure with the mean. Examples for correction: Arginine in Table 5 from 50.03 +/- 10.02 to 50 +/- 10, Arginine in Table 6 from 5.27 +/- 1.72 to 5.3 +/- 1.7
7) The bioinformatic tool MetOrigin and the literature should be used in order to clarify whether the metabolites reported in the study originate only from the microbiome, only from the human host, or have common origin (http://metorigin.met-bioinformatics.cn/.)
A native spaeker should review the manuscript.
Author Response
Dear Section Editor-in-Chief and dear Reviewers,
Thank you for taking of your time to evaluate our manuscript. Please find enclosed our revised version of the manuscript entitled “Quantitative, Targeted Analysis of Gut Microbiota Derived Metabolites Provides Novel Biomarkers of Early Diabetic Kidney Disease in Type 2 Diabetes Mellitus Patients” by Lavina Balint et al. for publication as a research article.
We have thoroughly revised the manuscript to meet the requirements of the reviewers, as it is described below:
Reviwer 3
- In table 2 the mean and SD for the control and CKD groups should be reported along with the ratio of mean CKD/mean control.
ANSWER: Thank you for your remark. We have performed additional statistical analysis and we have provided p-values resulted from the comparison between groups. Our statistical expert considers this provides sufficient data to demonstrate which molecules are potential biomarkers, as p-value is an optimal parameter.
2) In table 3 the units for LOD and LOQ should be reported
ANSWER: Checked.
Name |
Linear range (μM) |
Curve equation |
R2 |
LOD (μM) |
LOQ (μM) |
Arginine |
2-40 |
y=2476.6x+427.61 |
1 |
0.2 |
0.8 |
Hippuric acid |
5-200 |
y=2301.7x-1942.8 |
1 |
0.2 |
0.8 |
p-Cresylsulfate |
2.5-40 |
y=2717.3x-413.46 |
1 |
0.2 |
0.8 |
L-Acetylcarnitine |
1-5 |
y=36813x-3879.2 |
1 |
0.2 |
1.0 |
Indoxyl sulfate |
0.5-25 |
y=8157x-1240.8 |
1 |
0.2 |
0.8 |
Sorbitol |
0.2-4 |
y=39811x-1472.8 |
1 |
0.15 |
0.8 |
|
|
|
|
|
|
- For the recovery data presented in table 4 it is necessary to measure the recovery at 3 different concentrations for each metabolite (low, medium and high concentrations within the range of the standard curve. The concentration used for Hippuric acid is below the linear range so it should be at least 5 μM.
ANSWER: We did the recovery analysis for each metabolite with one concentration, indeed. Your observation is correct. For hippuric acid, we repeated the calibration curve, for the range 0.5 to 20 micromolar. The curve equation is corrected in Table 3. Thank you!
4) For the data presented in tables 5 and 6 it is necessary to perform statistical analysis in order to determine which metabolites exhibit significant differences between controls and CKD groups. It is also important to perform comparisons between the 3 CKD groups. Finally, the AUC for discriminating the different groups should be calculated for the most significantly different metabolites.
ANSWER: Table 5 and table 6 were tansformed into table 5 only, and then there was expressed the comparison between subgroups in serum, respectively in urine. We have performed additional statistical analyses, thus we have introduced new data in 3.2.2. subsection as it follows:
“3.2.2. Quantitative evaluation and statistical analysis
“With regard to statistical analysis, there were performed One Way ANOVA with Bonferroni correction, Chi squared test and Kruskal Wallis test, their results also being presented in table 5.”
Table 5. The mean values and standard deviations (±SD) of serum (mM) and urine concentrations (mM/mM creatinine) of the potential biomarkers targeted in this study for groups C and subgroups P1, P2 and P3.
BLOOD SERUM (μM) |
Control (n=20) |
P1 (n=29) |
P2 (n=29) |
P3 (n=32) |
Arginine |
50 (10) †,* |
44 (10.2)▲ |
39 (6.6) |
38 (7.8) |
Dimethyl Arginine |
0.9 (0.2) †,* |
1.1 (0.3) |
1.1 (0.5) |
1.2 (0.4) |
Hippuric acid |
24 (2.4) †⁑ |
22.7 (1.2) |
22.4 (1.7) |
21 (5) |
Indoxyl sulfate |
5 (0.5) * |
5.1 (0.5)▲ |
6.6 (5.2) |
6.6 (0.6)♦ |
L-Acetylcarnitine (AC) |
5.5 (2.1) |
5.7 (2.1) |
5.4 (1.7) |
5.6 (1.7) |
Butenoylcarnitine (eq AC) |
2.3 (0.1) * |
2.3 (0.1) ♣ |
2.6 (0.4) |
2.5 (0.5) |
Sorbitol |
2.5 (0.5) † |
2.3 (0.1) ♣ |
2.7 (0.3) |
2.6 (0.4) |
URINE (μM/μM creatinine) |
Control |
P1 |
P2 |
P3 |
Arginine |
5.3 (1.7) |
6.1 (2.9)▲ |
5 (3.4) |
5.7 (2.7) |
Dimethyl Arginine |
3.1 (0.4) ⁑ |
35.6 (7.5) |
21 (4.6) |
50.5 (11.1) |
Hippuric acid |
52.6 (29.4) |
54.7 (24.3) |
59.5 (73.3) |
59.5 (38.3) |
Indoxyl sulfate |
0.8 (0.4) ♦,* |
2.1 (1.1) |
2.1 (1.5) |
2.5 (1.2) |
p-cresylsulfate |
3.4 (1.6)†,⁑ |
5.7 (5.1)▲ |
5.9 (3.5) |
7.4 (6.4) |
L-Acetylcarnitine (AC) |
0.3 (0.1) |
0.4 (0.3) |
0.4 (0.3) |
0.6 (0.5) |
Butenoylcarnitine (eq AC) |
0.2 (0.1) ♦,* |
0.5 (0.2) |
0.5 (0.3) |
0.5 (0.3) |
Metabolites concentrations are presented as medians and standard deviations. Statistical significance between healthy controls and normoalbuminuric group, ♦ p<0.001; † p>0.001 and p<0.05; Statistical significance between normoalbuminuric group and microalbuminuric group, ♣ p < 0.001; ▲ p>0.001 and p<0.05; Statistical significance between microalbuminuric group and macroalbuminuric group, ♦ p<0.001; Statistical significance between healthy controls vs. normoalbuminuric group vs. microalbuminuric group vs. macroalbuminuric group; * p<0.001, ⁑ p>0.001 and p<0.05; P-values based on: *One way ANOVA with Bonferroni correction, **Chi squared test, ***Kruskal Wallis test
- It is important to compare the results of the untargeted analysis with the current targeted approach. The simplest way to do this comparison is to check the values determined by untargeted and targeted approaches of the ratio mean CKD/mean control for each metabolite.
ANSWER: Thank you for your suggestion. We did this, considering the MS peak intensities from the untargeted analysis (just for these biomarkers) and the values (expressed in concentrations) for the same targeted molecules. The values of these ratios were (more or less) similar.
6) The significant figures should be corrected throughout the manuscript. Please use two significant figures for the SD and then the mean should stop to the same significant figure with the mean. Examples for correction: Arginine in Table 5 from 50.03 +/- 10.02 to 50 +/- 10, Arginine in Table 6 from 5.27 +/- 1.72 to 5.3 +/- 1.7.
ANSWER: Thank you for your observation. We have performed the suggested corrections.
7) The bioinformatic tool MetOrigin and the literature should be used in order to clarify whether the metabolites reported in the study originate only from the microbiome, only from the human host, or have common origin (http://metorigin.met-bioinformatics.cn/.)
ANSWER: The MetOrigin is a comprehensive tool, very useful for asessing metabolite provenance. We have tried to perform an additional statistical analysis on this platform but we have encountered some limitations. In order achieve a Deep MetOrigin Analysis, there are required 3 tables that should encompass data about metabolites and microbiome. At this point, we do not own stool samples, in order to asess the microbiome of our subjects. Therefore, by the lack of this data, Deep MetOrigin Analysis cannot be performed. However, in our previous study (doi: 10.3390/ijms24076212), the gut origin of the metabolites was considered relevant according to literature data.
Thus, we have added an aditional statement regarding study limitations, namely: “Also, by the lack of stool sample analysis, our study can only point out the gut provenance of the metabolites according to literature data”
From this perspective, for the selection of the gut-derived metabolites, we have used literature data such as follows: Arginine – reference 31 – doi:10.3945/jn.115.226621, reference 32 – doi: 10.1016/j.phrs.2012.11.005; Uremic toxins (hippurate, indoxyl sulfate, p-cresul sulfate): New reference [13] - doi: 10.3390/ijms23010483, Acylcarnitines – reference 21 – doi:10.1016/j.ymgme.2015.09.004
Reviewer 4 Report
Dear Author,
Let me begin by congratulating you on excellent data and a well written article. The goal of this study was to identify novel biomarkers of early diabetic kidney disease in type 2 diabetes mellitus patients. You have successfully achieved this through the targeted analysis of selected metabolites previously identified using untargeted UHPLC-QTOF-ESI+-MS techniques. The discovery of a unique predictive signature profile is an important contribution to the field.
Although you have validated the results, it would be a sound idea, in future, to validate the results in an in vivo model, a preclinical study. Although not necessary to perform these studies for the publication of this particular manuscript, it would still be necessary to discuss preclinical papers that demonstrate how restoration of normal levels of these targeted predictive biomarker metabolites have a beneficial effect in preventing onset and progression of diabetic kidney disease in T2DM.
Once again, a very well written and relevant manuscript.
Author Response
Dear Section Editor-in-Chief and dear Reviewers,
Thank you for taking of your time to evaluate our manuscript. Please find enclosed our revised version of the manuscript entitled “Quantitative, Targeted Analysis of Gut Microbiota Derived Metabolites Provides Novel Biomarkers of Early Diabetic Kidney Disease in Type 2 Diabetes Mellitus Patients” by Lavina Balint et al. for publication as a research article.
We have thoroughly revised the manuscript to meet the requirements of the reviewers, as it is described below:
Reviwer 4
Thank you for appreciating our work and for offering us suggestions for future research directions.
Round 2
Reviewer 1 Report
line 86: you should add "(ADMA)" after "asymmetric dimethylarginine"
Table 3= Are you sure about Rsquare=1? Can you report the residual analysis? furthermore, linear regression does not analyse curves, it gives an equation of a straight line.
Which variable do you analyse through One-way ANOVA e which do you analyse through the Kruskas-Wallis test? yiu report "*One way ANOVA with Bonferroni correction, **Chi squared test, ***Kruskal Wallis test" but I do not see them in the table.
Author Response
Dear Section Editor-in-Chief and dear Reviewers,
Our team would like to thank you for revising our manuscript entitled “Quantitative, Targeted Analysis of Gut Microbiota Derived Metabolites Provides Novel Biomarkers of Early Diabetic Kidney Disease in Type 2 Diabetes Mellitus Patients” by Lavinia Balint et al. Please find attached the revised manuscript and revised supplementary materials, which we have submitted for publication.
Your valuable suggestions have a major contribution in improving our manuscript, thus we have treated them accordingly.
Reviewer 1
Thank you for your pertinent observations.
- line 86: you should add "(ADMA)" after "asymmetric dimethylarginine"
ANSWER: We had adjusted the term “ADMA” according to your request.
- Table 3= Are you sure about Rsquare=1? Can you report the residual analysis? furthermore, linear regression does not analyse curves, it gives an equation of a straight line
ANSWER: In the first round of reviews, one of the reviewers had the following request:
“The significant figures should be corrected throughout the manuscript. Please use two significant figures for the SD and then the mean should stop to the same significant figure with the mean. Examples for correction: Arginine in Table 5 from 50.03 +/- 10.02 to 50 +/- 10, Arginine in Table 6 from 5.27 +/- 1.72 to 5.3 +/- 1.7”
Thus, we have corrected the entire manuscript according to the reviewer’s request. We are aware of the fact that R squared can rarely reach the absolute value = 1. Therefore, we have provided the residual analysis, with the exact R2 values in table 3, in order to avoid any confusion.
Table 3. Validation parameters (linear range, curve equation, R2, LOD and LOQ) for each of the seven molecules selected as potential biomarkers.
Name |
Linear range (μM) |
Curve equation |
R2 |
LOD (μM) |
LOQ (μM) |
Arginine |
2-40 |
y=2476.6x+427.61 |
0.999 |
0.2 |
0.8 |
Hippuric acid |
0.5-10 |
y=5097.7x-441.68 |
0.999 |
0.2 |
0.8 |
p-Cresylsulfate |
2.5-40 |
y=2717.3x-413.46 |
0.999 |
0.2 |
0.8 |
L-Acetylcarnitine |
1-5 |
y=36813x-3879.2 |
0.994 |
0.2 |
1.0 |
Indoxyl sulfate |
0.5-25 |
y=8157x-1240.8 |
0.999 |
0.2 |
0.8 |
Sorbitol |
0.2-4 |
y=39811x-1472.8 |
0.999 |
0.15 |
0.8 |
- Which variable do you analyse through One-way ANOVA e which do you analyse through the Kruskas-Wallis test? yiu report "*One way ANOVA with Bonferroni correction, **Chi squared test, ***Kruskal Wallis test" but I do not see them in the table.
ANSWER: If you may refer to table 1, subjects enrolled and HbA1c were compared and analysed between subgroups by One Way ANOVA (marked after the p-value with “ * ”), sex (female), diabetic polineuropathy, diabetic retinopathy were assesed by chi squared test (marked after the p-value with “ ** ”), and DM duration and uACR were evaluated by Kruskall Wallis test (markerd after the p-value with (“ *** “). In addition, starting from line 343, the paragraph regarding statistical analyses was redrafted as it follows:
“The statistical analysis was performed by: (1) One Way ANOVA with Bonferroni correction analysis, Chi squared test and Kruskal Wallis test – which allowed for the differentiation of clinico-biological features between subgroups as presented in table 1; and (2) Mann-Whitney test which permited the evaluation of metabolite differences between subgroups – data presented in table 5.”
Reviewer 3 Report
The authors have addressed most of my comments. The following revisions are required
Revision 1
- In table 2 the mean and SD for the control and CKD groups should be reported along with the ratio of mean CKD/mean control.
ANSWER: Thank you for your remark. We have performed additional statistical analysis and we have provided p-values resulted from the comparison between groups. Our statistical expert considers this provides sufficient data to demonstrate which molecules are potential biomarkers, as p-value is an optimal parameter.
Reviewer: The mean and standard deviation for all the metabolites in each group should be reported in Table 2. This is a requirement for clear data presentation and has nothing to do with statistics. The data are available and should be presented.
Revision 2
- For the recovery data presented in table 4 it is necessary to measure the recovery at 3 different concentrations for each metabolite (low, medium and high concentrations within the range of the standard curve. The concentration used for Hippuric acid is below the linear range so it should be at least 5 μM.
ANSWER: We did the recovery analysis for each metabolite with one concentration, indeed. Your observation is correct. For hippuric acid, we repeated the calibration curve, for the range 0.5 to 20 micromolar. The curve equation is corrected in Table 3. Thank you!
Reviewer: The quality of the publication will be higher if the recovery analysis is performed at 3 concentrations (low, medium, high) but I will not insist if the authors choose to omit this analysis.
The calibration curve for hippuric acid reported in Table 3 and Fig. S1. is still for the range of 5-200 micromolar. The new data from the standard curve with range 0.5 to 20 micromolar should be reported.
Revision 3
- It is important to compare the results of the untargeted analysis with the current targeted approach. The simplest way to do this comparison is to check the values determined by untargeted and targeted approaches of the ratio mean CKD/mean control for each metabolite.
ANSWER: Thank you for your suggestion. We did this, considering the MS peak intensities from the untargeted analysis (just for these biomarkers) and the values (expressed in concentrations) for the same targeted molecules. The values of these ratios were (more or less) similar.
Reviewer: These data should be reported in a supplementary table
Author Response
Dear Section Editor-in-Chief and dear Reviewers,
Our team would like to thank you for revising our manuscript entitled “Quantitative, Targeted Analysis of Gut Microbiota Derived Metabolites Provides Novel Biomarkers of Early Diabetic Kidney Disease in Type 2 Diabetes Mellitus Patients” by Lavinia Balint et al. Please find attached the revised manuscript and revised supplementary materials, which we have submitted for publication.
Your valuable suggestions have a major contribution in improving our manuscript, thus we have treated them accordingly.
Reviewer 3
Thank you for your answer and your fair observations.
- The mean and standard deviation for all the metabolites in each group should be reported in Table 2. This is a requirement for clear data presentation and has nothing to do with statistics. The data are available and should be presented.
ANSWER: In order to acomplish this, we have enlarged the table 2 and we have introduced the means of peak intensities (PI) in DKD group, C group, and the ratio of mean PI DKD/mean PI C. Starting from line 282 we have introduced the following additional data:
“In addition, table 2 displays the mean peak intesities (PI) of DKD group and of C group, and the ratio of mean DKD/mean C group.
Table 2. The classification of metabolites that discriminate the group DKD from group C, based on mean PI in DKD group, mean PI in C group, ratio DKD/C, RT, m/z and AUC values in serum and urine.
Serum |
||||||||||||
m/z |
|
Identification |
PI DKD group |
PI C group |
Ratio DKD/C |
RT (min) |
AUC |
|||||
Mean |
±SD |
Mean |
±SD |
|||||||||
175.1306 |
Arginine |
100039.21 |
47525.72 |
124331.91 |
25243.14 |
0.80 |
1 |
0.5 |
||||
180.1716 |
Hippuric acid |
111747.94 |
46825.82 |
123619.30 |
11537.92 |
0.90 |
11 |
0.7 |
||||
183.0940 |
Sorbitol |
98996.89 |
37792.49 |
99647.14 |
16840.26 |
0.99 |
1 |
0.5 |
||||
204.1369 |
L-Acetylcarnitine |
201181.48 |
108608.92 |
199328.56 |
74900.62 |
1.01 |
1 |
0.6 |
||||
214.2676 |
Indoxyl sulfate |
48652.85 |
17782.53 |
40115.19 |
2429.85 |
1.21 |
12 |
0.6 |
||||
230.2668 |
Butenoyl carnitine |
87016.09 |
34019.37 |
80790.60 |
170.13 |
1.08 |
11 |
0.6 |
||||
Urine |
||||||||||||
m/z |
Identification |
PI DKD group |
PI C group |
Ratio DKD/C |
RT (min) |
AUC |
||||||
Mean |
±SD |
Mean |
±SD |
|||||||||
175.1306 |
Arginine |
14288.87 |
7608.37 |
13479.29 |
4687.36 |
1.06 |
1 |
0.7 |
||||
180.1716 |
Hippuric acid |
294865.25 |
229329.22 |
267442.47 |
149277.77 |
1.10 |
11 |
- |
||||
204.1369 |
L-Acetylcarnitine |
16892.46 |
13440.22 |
7514.83 |
1256.62 |
2.25 |
1 |
0.5 |
||||
214.2676 |
Indoxyl sulfate |
16693.72 |
9049.71 |
5529.51 |
1777.29 |
3.02 |
12 |
1 |
||||
189.1594 |
p-Cresylsulfate |
16814.83 |
13305.89 |
8798.191 |
4015.74 |
1.91 |
1 |
0.8 |
||||
230.2668 |
Butenoyl carnitine |
17342.30 |
9668.83 |
4955.82 |
170.13 |
3.50 |
11 |
1 |
||||
* Data presented as means ±SD; AUC – area under curve; m/z – mass to charge ratio; PI – peak intesity; RT – retention time;
- The calibration curve for hippuric acid reported in Table 3 and Fig. S1. is still for the range of 5-200 micromolar. The new data from the standard curve with range 0.5 to 20 micromolar should be reported.
ANSWER: We apologize for this, an error might have had occurred. We have corrected hippuric acid calibration curve in Fig.S2 and table 3, respectively.
- It is important to compare the results of the untargeted analysis with the current targeted The simplest way to do this comparison is to check the values determined by untargeted and targeted approaches of the ratio mean CKD/mean control for each metabolite. These data should be reported in a supplementary table.
ANSWER: In order to adress this issue, we have submitted the requested Supplementary file S3 and, in line 342, it was introduced the following phrase: “The calibration curves are presented in Supplementary file 2 and the comparison between untargeted and targeted analyses of the metabolites expressed in mean DKD/mean C ratios, are given in Supplementary file 3.”